# Interactive Segmentation with Elaborate Focus Prior

**Kangpeng Hu** [1]  **Yinghui Sun** [2]  **Tao wang** [1]  **Weihao Zhang** [1]  **Quansen Sun** [1]

## Abstract

Regional refinement for interactive segmentation is of great necessity to ensure the fidelity of segmented pixels nearby user-prompted locations, which specifies a local window (i.e., focus view) for the latest click after a global prediction, where local pixels are revisited and optimized using numerous refining structures. Previous methods either utilize a two-stage pipeline to estimate the focus view or manually preset a fixed scope for all clicks, while the former is time-consuming, the latter fails to capture the correlation among click position, object geometry, and focus intensity. In this paper, we inherit the core idea of FCFI (Wei et al., 2023) and dedicate a one-stage framework characterized with **E**laborate **F**ocus **P**rior (EFPNet). Concretely, EFPNet outputs an erroneous mask *w.r.t* historical feedback and newly-placed click in an end-to-end manner, which deduces precise focus region according to its max-connected component, followed with feedback correction considering image, feature and mask affinity. We further design a clicked-with-focus mechanism for delicate feedback integration. Extensive studies on four benchmarks have revealed its outstanding performance for both efficacy and efficiency.

## 1. Introduction

Interactive segmentation (IS) aims to localize salient object region guided by various form of prompts, including clicks, texts and audios, which indicates specific user intentions. Until a best result is returned, user is allowed to alternately provide positive or negative prompts in order to erase the imperfections such as holes and overstuffed pixels. Due to its flexibility of segmenting arbitrary objects, it has been harvesting massive promotion in domains such as data annotation and pathological analysis.

---

[1]Nanjing University of Science and Technology, Nanjing, China [2]Southest University, Nanjing, China. Correspondence to: Yinghui Sun <sunyh@seu.edu.cn>.

*Proceedings of the $43^{rd}$ International Conference on Machine Learning*, Seoul, South Korea. PMLR 306, 2026. Copyright 2026

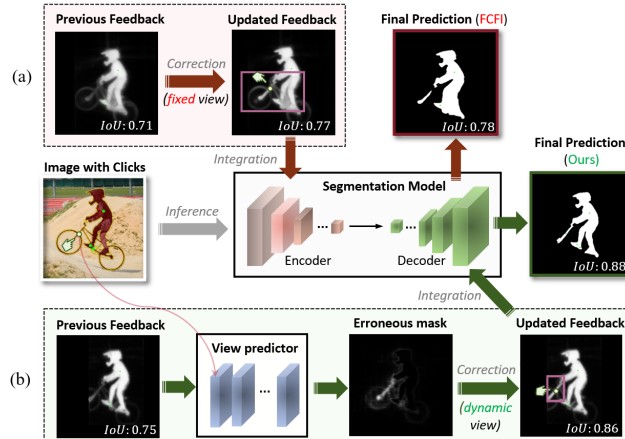

*Figure 1.* (**a**): Pipeline for FCFI, which leverages solid focus view during feedback correction. (**b**): Pipeline for our proposed EFPNet, which obtains dynamic focus view based on end-to-end prediction of erroneous region.

Numerous attempts have been made in the past decades to exploit the performance of IS from multifaceted aspects, including network design (Lin et al., 2020; Chen et al., 2021; Huang et al., 2023; 2024b; 2025), prompt encoding (Maninis et al., 2018; Kirillov et al., 2023; Popenova et al., 2023), training strategy (Sofiiuk et al., 2022; Sun et al., 2023; Antonov et al., 2025) and post-processing (Sofiiuk et al., 2020; Chen et al., 2022; Wei et al., 2023; Choi et al., 2024; Lee et al., 2024), etc. In terms of simplicity, click has been widely adopted as the major interaction tool in most methods. DIOS (Xu et al., 2016) is the pioneering technique that regulates the basic pipeline of click-based IS: Given a target image and one positive click, a deep network encodes them into fused features and decodes a primary mask, then an extra click is added to network input so as to revise the mislabeled regions. The above procedure is back-and-forth until the result is satisfactory.

Meanwhile, previous works also strive to boost the prediction accuracy of local patches centered around clicks. They notice that the segmentation feedback of click-marked pixels may be inconsistent with user preference, say, region placed with positive click is inferred as negative, or vice versa. This is probably due to the sparsity of click

prior which suffers from feature dilution, especially when network goes deeper. A non-trivial solution is to define a focus view *w.r.t.* click position, while feedback within this view will undergo several post-processing steps to get further refined. For instance, FocusCut (Lin et al., 2022) and FocalClick (Chen et al., 2022) estimate the focus view by measuring the subtracted area between current prediction and previous feedback, then the focus region is cropped and revised by an extra network in a two-stage manner. In contrast, FCFI (Wei et al., 2023) presets focus view with a fixed ratio of 0.3, then performs feedback correction and integration based on feature affinity and channel-wise fusion. The former case could precisely delineate the range of focus yet brings about redundant computation. While the latter case is time-efficient, its lack of focus specificity is prone to detail leakage when the deviation of focused region *w.r.t.* mislabeled area is critical, causing poor refinement quality.

In this paper, we further investigate how to make full use of focus prior during IS process while pursuing a trade-off between quality and cost. Our work is a well-improved version of FCFI (Wei et al., 2023), since it is the current SOTA method with regard to local refinement scheme. In our proposed EFPNet, we strive to tackle four drawbacks of the raw FCFI pipeline: (*i*) Focus region is constant wherever click is located. (*ii*) Local refinement is not available when user clicks for the first time. (*iii*) Feature-wise affinity is an incomplete prior for feedback correction. (*iv*) Convolution-based feedback integration is prone to unstable mask output. For *i* and *ii*, we start from a *learnable* focus view predictor (FVP) which produces an erroneous mask that highlights the wrongly-classified pixels in previous feedback, the focus view is then deduced according to its max-connected region around the click. In one-click scenario where previous feedback is an all-zero mask, FVP is equivalent to full-object segmentation task. We further complement the feedback correction process with three types of affinity (TAC), which resolves *iii*. To tackle *iv*, we propose a clicked-with-focus integration module (CFI) including inter-focus contrastive learning and intra-focus attention. Evaluation on four datasets (*i.e.*, GrabCut (Rother et al., 2004), Berkeley (Martin et al., 2001), DAVIS (Perazzi et al., 2016), SBD (Hariharan et al., 2011)) reveals a remarkable breakthrough against other methods.

## 2. Related Work

**Interactive image segmentation**. As a long-standing research topic, interactive segmentation (IS) accounts for prompt instruction from labor to produce specific and class-agnostic instance masks. In the early period, graph-based or energy-based method is popular, such as GraphCut (Boykov & Jolly, 2001), random walk (Grady, 2006) and active contour (Kass et al., 1988), which lacks accuracy when handling complex scenes. As the first deep-learning-based method, DIOS (Xu et al., 2016) encodes clicks into 2-channel map and concatenate with input image, which are sent to FCN (Long et al., 2015) to yield a binary mask. They also explore the clicking behavior of real-world users and propose a novel click simulation pipeline for training. Based on this paradigm, numerous works have been raised in which they delve into various perspectives, such as powerful feature extractor (Li et al., 2023; Chen & Zhao, 2025), prompt modalities (Kirillov et al., 2023; Popenova et al., 2023; Chen et al., 2023; Antonov et al., 2025), training strategy (Liu et al., 2022; Sun et al., 2023), granularity (Li et al., 2018; Liew et al., 2019; Zhao et al., 2024; Ravi et al., 2024; Huang et al., 2024a), post-processing (Liew et al., 2017; Chen et al., 2022; Wei et al., 2023; Sofiiuk et al., 2020; Zhou et al., 2023; Choi et al., 2024), and attention-based design (Chen et al., 2021; Liu et al., 2023; Huang et al., 2023; 2024b; 2025).

**Regional refinement for interactive segmentation**. As a pioneering work for two-stage refinement, RIS-Net (Liew et al., 2017) uses a parallel branch to generate local predictions within region-of-interests (rois) for all clicks and merge with global prediction, where the rois are restricted by minimal distance within click pairs. Later, BRS (Jang & Kim, 2019) and f-BRS (Sofiiuk et al., 2020) propose a back-propagation refinement scheme which fine-tunes network input. Moreover, f-BRS crops and infers the salient object region of previous feedback using Zoom-In strategy. Similar to RIS-Net, FocalClick (Chen et al., 2022) and FocusCut (Lin et al., 2022) adopts a coarse-to-fine pipeline attached with auxiliary refining networks. After the global prediction is obtained, it crops a local patch from image, click map and predicted mask to feed the refining network and paste the local prediction back to global. As a novel one-stage method, FCFI (Wei et al., 2023) modifies the previous feedback within a static focus scope around the click based on feature affinity, acting as a dense prior to enhance the perception of focused region.

## 3. Method

### 3.1. Preliminary

An overview of our proposed EFPNet is illustrated in Figure 2. Before depicting our work, it is necessary to revisit the design pattern of FCFI (Wei et al., 2023) in below.

**Interaction Setup**. Given an input image $I \in \mathbb{R}^{H \times W \times 3}$, user is required to place $N$ positive or negative clicks on image canvas, denoted as $\mathcal{C} = \{(u_i, v_i, p_i)\}_{i=1}^{N}$, where $(u_i, v_i) \in [0, W] \times [0, H]$ and $p_i \in \{0, 1\}$ indicate the coordinate and property (*i.e.*, positive for 1 and negative for 0) of the $i$-th click, respectively. The clicks are encoded into a 2-channel disk map $S \in \mathbb{R}^{H \times W \times 2}$ to concatenate with image $I$, where a pretrained backbone is leveraged to extract

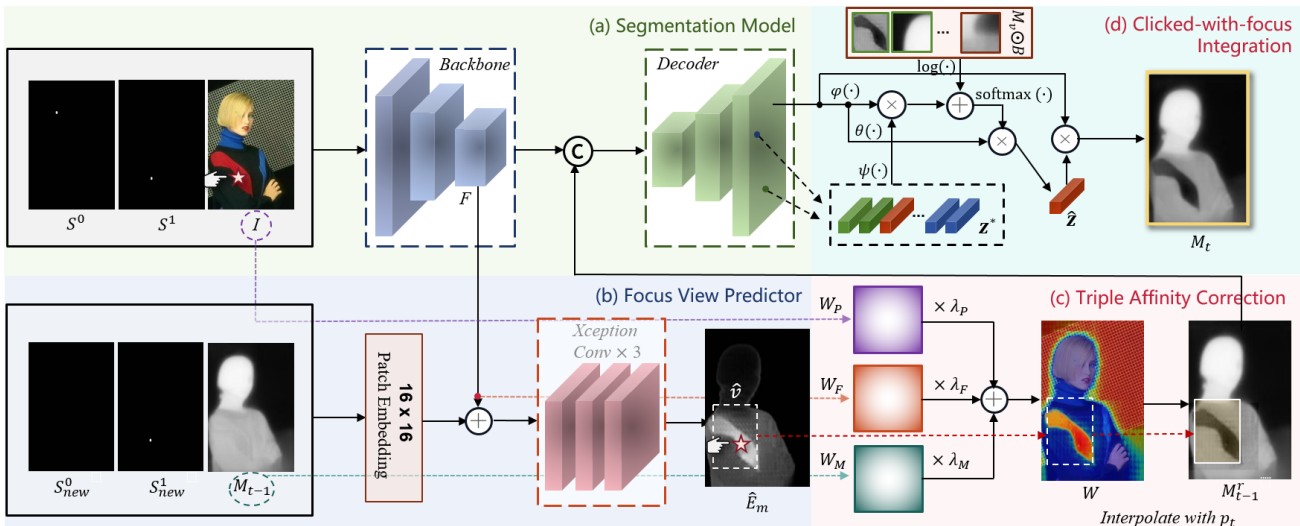

**Figure 2.** Our EFPNet framework. Given an input image $I$, its backbone feature $F$ will be first leveraged for focus prediction, then previous feedback $M_{t-1}$ within this focus will be refined as $M_{t-1}^r$ based on triple affinity, which is integrated with $F$ to yield final $M_t$.

feature $F \in \mathbb{R}^{\frac{H}{16} \times \frac{W}{16} \times d}$ from the 5-channel input, then $F$ is inferred using a decoder network to form a prediction mask $M_t \in [0, 1]^{H \times W}$, where $t$ is the round of interaction.

**Feedback Correction**. Some works (Liu et al., 2023) revise the function of previous feedback $M_{t-1}$ as additional input channel along with image $I$ and clicks $\mathcal{C}$ to fully exploit the prior information. In FCFI, a *Focused Feedback Correction Module* (FFCM) takes charge of refining $M_{t-1}$ in a local patch $v$ centered with the new click $c_{new} = (u_t, v_t, p_t) \in \mathcal{C}$ by calculating the affinity $W_F$ between $F(u_t, v_t)$ and $F$. The shape of $v$ is fixed on $0.3H \times 0.3W$. For each pixel in $M_{t-1}$, its value is interpolated with label $p_t$ based on $W_F$ to obtain the refined feedback $M_{t-1}^r$. This process will be ignored when there's only one click.

**Feedback Integration**. To avoid prior dilution, FCFI leverages a *Collaborative Feature Fusion Module* (CFFM) to integrate the corrected feedback with intermediate feature instead of early-fusion. After concatenating $F$ with $M_{t-1}^r$, a few convolution layers with residual connection are presented for global refinement and feature enhancement, which returns a fused feature $F_u$ for segmentation.

### 3.2. Focus View Predictor

As a latent property, the intensity of prior diffusion for each click is restricted to a certain scope, which constitutes a global user intention with the participation of all clicks. In classical methods, this intensity is a vaguely-estimated measurement which is *implicitly* perceived during feature propagation based on appearance and semantic homogeneity in the proximity of click. However, a scene may suffer from complex object shapes, occlusions, and severe lighting conditions, which brings ambiguity and uncertainty such

that *explicitly* reasoning this intensity becomes necessary.

To handle this, recent works (Lin et al., 2022; Chen et al., 2022) adopts the concept of *focus view* which enforces network to locally refine the focused region while neglecting others. While FCFI (Wei et al., 2023) performs feature-wise crop-and-paste to reduce inference cost in (Lin et al., 2022), it is incapable of completely covering the mislabeled region under extreme cases such as thin object (*e.g.*, pipes), irregular object (*e.g.*, clouds) and cluster(*e.g.*, a group of birds), which results in detail leakage.

Therefore, we propose an end-to-end *focus view predictor* (FVP). We notice that during the training stage, two erroneous parts of previous feedback (*w.r.t.* ground truth $G$) are calculated (denoted as $E \in \{E_p, E_n\}$), where $E_p$ means false positive and $E_n$ means false negative. A new click $c_{new}$ is then randomly sampled in either $E_p$ or $E_n$ based on their pixel summation (denoted as $O_p$ and $O_n$): if $O_p$ is larger than $O_n$, then a negative click is chosen from $E_p$, otherwise a positive click is chosen from $E_n$. This process is formulated as follows:

$$E_p = M_{t-1} \circledast (1 - G), E_n = (1 - M_{t-1}) \circledast G \quad (1)$$

$$c_{new} \in \{(x, y, p) | E(x, y) = 1\}_{p = \arg\max(O_p, O_n)} \quad (2)$$

where $\circledast$ is the AND operation. For $c_{new}$, it takes charge of modifying the pixels in $E$ within a certain extent—only the max-connected region involving $c_{new}$ (denoted as $E_m$) is kept while abolishing other isolated regions.

Up to now, we're able to deduce the ideal focus view (denoted as $v \in \mathbb{R}^4$) for $c_{new}$ by computing the tightest bounding box of region $E_m$. Apparently, $E_m$ heavily relies on ground-truth $G$, which is infeasible in real-world scenario.

However, we argue that $E_m$ could also be a **learnable** posterior term *w.r.t.* $c_{new}$ and $M_{t-1}$, since during the inference stage, $c_{new}$ is freely set by user with no priori restriction from $G$, thus we could construct a mapping in below:

$$\hat{E}_m = f(I, M_{t-1}, c_{new}; \theta_f) \tag{3}$$

where $f$ is an arbitrary deep network which parses the prior from image $I$, previous feedback $M_{t-1}$ and the new click $c_{new}$ collectively to predict the error mask $\hat{E}_m$, which indicates the expected focus objective in round $t$. In practice, we introduce a lightweight decoder consisting of three Xception (Chollet, 2017) blocks and a classifier conv (denoted as $\phi_d(\cdot)$), which takes backbone feature $F$ as input since $F$ is naively derived from $I$ and $\mathcal{C}$, where $c_{new} \in \mathcal{C}$. However, we argue that the encoding function of $\mathcal{C}$ (*i.e.*, disk map) treats all clicks equally hence eliminates temporal (round of click) information. Therefore, we encode $c_{new}$ into a separate map $S_{new} \in \mathbb{R}^{H \times W \times 2}$ and concatenate with $M_{t-1}$, while a patch embedding layer (Liu et al., 2021) (denoted as $\phi_p(\cdot)$) downsamples the 3-channel input into a 16x prior feature to align and fuse with $F$. We keep the all-zero channel in $S_{new}$ to retain the property for $c_{new}$, which proves essential to identify the expected type of error ($E_p$ or $E_n$). We formulate the above process as follows:

$$\hat{E}_m = \phi_d(F \oplus \phi_p(M_{t-1}, S_{new}; \theta_p); \theta_d) \tag{4}$$

$$\hat{v} = \text{BBox}(\hat{E}_m > \tau) = \{t_m, l_m, b_m, r_m\} \tag{5}$$

where $\oplus$ is element-wise addition, $\tau$ is a binary threshold to control the range of predicted focus view $\hat{v}$. During training, $\hat{E}_m$ is supervised by $E_m$ using focal loss (Lin et al., 2017). Note in the primitive round ($t = 1$), user is inclined to place a positive click $c_0$ in the center of object, which stands for *global view*. We notice that our method is also feasible for global view prediction, since the view for $c_0$ is required to overlap the whole object, namely $E_m = E = G$. In this scenario, FVP is converted to a *instance segmentation task*, manifesting primary user intention rather than detailed error.

### 3.3. Triple-Affinity Correction

Given the predicted $\hat{v}$, previous feedback $M_{t-1}$ will be partially refined as $M_{t-1}^r$. We start from measuring the normalized feature affinity $W_F \in [0, 1]^{\frac{H}{16} \times \frac{W}{16}}$ between $c_{new} = \{u_t, v_t, p_t\}$ and other pixels in backbone feature $F$:

$$W_F(i, j) = 0.5 \times \frac{F(u_t, v_t) \cdot F(i, j)}{||F(u_t, v_t)||_2 ||F(i, j)||_2} + 0.5 \tag{6}$$

Then, an indicator mask $M_v \in \{0, 1\}^{\frac{H}{16} \times \frac{W}{16}}$ is initialized with pixels in the scope of $\hat{v}$ set to 1 while others set to 0. Based on Section 3.1, $M_{t-1}^r$ is derived by interpolating $M_{t-1}$ for twice, which can be written as follows:

$$M_{t-1}^o = p_t \cdot W_F + (1 - W_F) \odot M_{t-1} \tag{7}$$

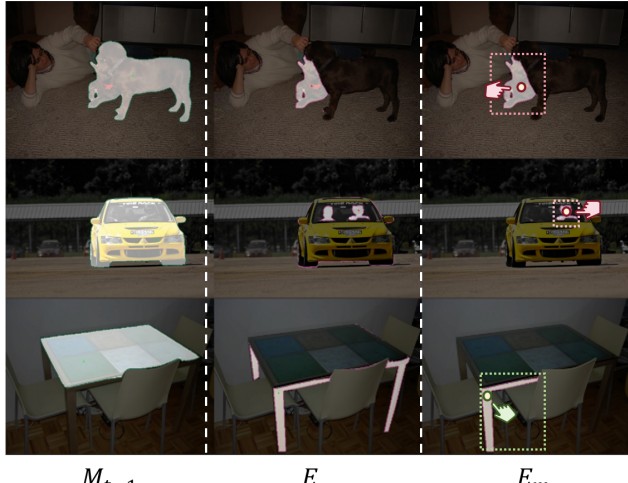

*Figure 3.* Examples for click-centered erroneous mask $E_m$ with focus view. The click is marked with red and green to indicate false positive $E_p$ and false negative $E_n$.

$$M_{t-1}^r = M_{t-1}^o \odot M_v + (1 - M_v) \odot M_{t-1} \tag{8}$$

where $\odot$ means Hadamard product. As a training objective, $M_{t-1}^r$ is supervised with ground truth $G$, which is in essence a region-based constrastive learning process for $F$ that pulls closer the similar pairs of feature and pushes away others. However, we notice that simply leveraging $W_F$ for refinement is less error-tolerance, since $F$ is generated from a deep backbone where prior dilution occurs, leaving incomplete optimization during training.

Therefore, we search for other candidate modalities that provide precise and stable prior guidance to $M_{t-1}^r$, which forms our *Triple-affinity Correction* (TAC) module. As a low-level prior, the input image $I$ is enriched with visual details (*e.g.*, color, texture) which fades in $F$ due to network depth. We then propose a pixel affinity $W_P$ to complement with $W_F$, where softmax is performed in the scope of $\hat{v}$:

$$W_P(i, j) = \frac{exp(\kappa_p(I_{i,j}, I_{u_t, v_t}))}{\sum_{(s,t) \in \hat{v}} exp(\kappa_p(I_{s,t}, I_{u_t, v_t}))} \tag{9}$$

$$\kappa_p(I_{i,j}, I_{u_t, v_t}) = -\frac{|I(i, j) - I(u_t, v_t)|}{\sigma_p^2} \tag{10}$$

On the other hand, revisiting $M_{t-1}$ itself could also bring about some special observations. For example, background pixel in zone $[0, 0.5)$ could be further partitioned into multiple sub-regions—when infinitely close to 0, it belongs to wide and empty stuff (*i.e.*, sky, road); when infinitely close to 0.5, it indicates that there exists latent object (or part of object) with clear semantic, since the *uncertainty* of pixel classifier has reached a climax. Meanwhile, we find that sub-region (whatever object or stuff) shares consistent pixel values with small variation, which sheds light on our

proposed mask affinity $W_M$ in below:

$$W_M(i,j) = (1 - \kappa_m(M_{t-1}^{i,j}, M_{t-1}^{u_t,v_t}))^\gamma \quad (11)$$

$$\kappa_m(M_{t-1}^{i,j}, M_{t-1}^{u_t,v_t}) = max(|M_{t-1}(i,j) - 0.5| \\ , |M_{t-1}(i,j) - M_{t-1}(u_t,v_t)|) \quad (12)$$

where $|M_{t-1}(i,j) - 0.5|$ acts as a penalty to prevent user from erroneously setting a positive click on stuff from the second round (if user is intended to segment the object region), or vice versa. We set $\gamma = 3$ by default.

Finally, we merge them into a unified affinity form $W = \lambda_P W_P + \lambda_F W_F + \lambda_M W_M$ to replace $W_F$ in Equation (7), where different combination of weighting term $\lambda_P$, $\lambda_F$ and $\lambda_M$ will be further discussed in Section 4.3.

### 3.4. Clicked-with-Focus Integration

As shown in Section 3.1, feedback integration process in FCFI is just a stack of convolution layers. This design is prone to severely unstable output, mainly because (*i*) It reveals a low utilization rate of focus prior; (*ii*) There's no guarantee that the corrected focus details of each historical round could be inherited in the new round. To handle these issues, we propose *Clicked-with-Focus Integration* (CFI), which consists of *Inter-Focus Constrastive Learning* (FCL) and *Intra-Focus attention* (FA) components.

**Inter-Focus Constrastive Learning**. In addition to newly-placed click, we maintain a list to record the focus view for $t-1$ rounds of historical click, denoted as $\{\hat{v}_i\}_{i=1}^t$. Given corrected feedback $M_{t-1}^r$, we compute the focused feature $\{F_i^l\}_{i=1}^t$ for each click $c \in \{(u_i, v_i, p_i)\}_{i=1}^t$ in below:

$$F_i^l = \frac{1}{|M_{v_i} \odot B_i|} \sum_p F(p) M_{v_i}(p) B_i(p) \quad (13)$$

$$B_i = M_{t-1}^r \text{ if } p_i = 1 \text{ else } 1 - M_{t-1}^r \quad (14)$$

Then, we construct a cosine-similarity matrix for each pair of feature vector in $\{F_i^l\}_{i=1}^t$, where features with the same click property should be closer to each other, otherwise pushed away. We strive to optimize the matrix elements selectively, where we pick out $K_p$ positive samples with low scores and $K_n$ negative samples with high scores. Then, we propose a regularization term $L_{fcl}$ as follows:

$$L_{fcl} = \frac{1}{K_n} \sum_{c_i,c_j}^{\mathcal{C}} \mathbb{I}[p_i \neq p_j] \mathbb{I}[S(c_i,c_j) \geq \tau_n] S(c_i,c_j) \\ - \frac{1}{K_p} \sum_{c_i,c_j}^{\mathcal{C}} \mathbb{I}[p_i = p_j] \mathbb{I}[S(c_i,c_j) \leq \tau_p] S(c_i,c_j)$$

$$(15)$$

where $\mathbb{I}[\cdot]$ is indicator function, $S(c_i,c_j) = \langle F_i^l, F_j^l \rangle$ is cosine similarity, $\tau_p = 0.75$ and $\tau_n = 0.5$ are positive and negative threshold. This design brings about a powerful constraint for the stability of backbone feature $F$, which ensures that the refinement prior of historical clicks will be inherited and interact with each other so as to avoid feature collapse in the well-refined regions.

**Intra-Focus Attention**. We start from an arbitrary decoder which takes $F$, $M_{t-1}^r$ as input and output $F_d \in \mathbb{R}^{\frac{H}{4} \times \frac{W}{4} \times d}$. We then accumulate all pixels sampled with $\mathcal{C}$ to obtain the click embedding $F_c = \{f_i | f_i = F_d(c_i)\}_{i=1}^t$.

Next, we initialize a learnable mask token $\mathbf{z} \in \mathbb{R}^d$ to concatenate with $F_c$ in the primary axis, which forms $\mathbf{z}^* \in \mathbb{R}^{(t+1) \times d}$. Inspired by (Ravi et al., 2024), we establish the feature pathway between click-to-click, click-to-focus, local-to-global, and prior-to-token in a collective way, using focus-oriented attention mechanism as follows:

$$\tilde{\mathbf{z}}^* = \mathbf{z}^* + \text{softmax}(\frac{QK^T}{\sqrt{d_k}} + \log(M_v \odot B))V \quad (16)$$

where $Q = \varphi(\mathbf{z}^*), K = \psi(F_d), V = \theta(F_d)$. $B = \{B_i\}_{i=1}^t$ follows the definition in Equation (14). The $\log(\cdot)$ enforces the attention value outside $\{\hat{v}_i\}_{i=1}^t$ to be infinitesimal so that the perception range for each click is limited within the focus except for token $\mathbf{z}$, whose attention value is all-one for global view. We then perform self-attention for $\tilde{\mathbf{z}}^*$ to mutually propagate from clicks to token, which yields $\hat{\mathbf{z}}^*$.

Lastly, the condensed token $\hat{\mathbf{z}}$ will be extracted from $\hat{\mathbf{z}}^*$ and convolve with $F_d$ to obtain final prediction $M_t$. In essence, it imitates the manner of a standard SAM decoder layer (Kirillov et al., 2023) but with designated attention scope for each of its mask and click token, since each click is expected to gather image information in its *vicinity* with *shared property*, while mask token takes charge of accumulating the local feedback of all clicks. We show in our ablation study that this design critically avoids the redundancy and perturbation from irrelevant pixels.

## 4. Experiment

### 4.1. Experimental Settings

**Datasets.** Following (Wei et al., 2023; Zhao et al., 2024; Lin et al., 2025), we adopt SBD (Hariharan et al., 2011), COCO (Lin et al., 2014) and LVIS (Gupta et al., 2019) for training and GrabCut (Rother et al., 2004), Berkeley (Martin et al., 2001), DAVIS (Perazzi et al., 2016), SBD (Hariharan et al., 2011) for evaluation.

**Evaluation Protocols.** We inherit commonly-used testing metrics from previous works, including a) Number of clicks (NoC%x), the least number of clicks to reach a given IoU threshold $x$. b) Number of failure (NoF%x), cases that fail to reach %x IoU within 20 clicks. c) Params, FLOPs, and Second per click (SPC), for efficiency analysis. d) $x$-IoU,

*Table 1.* Comparison between our proposed EFPNet and other mainstream methods. The 1st/2nd best result is marked with red/light red (trained with COCO+LVIS) and green/light green (trained with SBD), respectively.

| Method | Backbone | Train set | GrabCut | | Berkeley | | DAVIS | | SBD | |
|---|---|---|---|---|---|---|---|---|---|---|
| | | | NoC%85 | NoC%90 | NoC%85 | NoC%90 | NoC%85 | NoC%90 | NoC%85 | NoC%90 |
| *No Local Refinement* | | | | | | | | | | |
| CDNet (Chen et al., 2021) | ResNet-101 | SBD | 2.42 | 2.76 | - | 3.65 | 5.33 | 6.97 | 4.73 | 7.66 |
| RITM (Sofiiuk et al., 2022) | HRNet-18s | SBD | 1.76 | 2.04 | 1.87 | 3.22 | 4.94 | 6.71 | 3.39 | 5.45 |
| SimpleClick (Liu et al., 2023) | ViT-B | SBD | 1.40 | 1.54 | 1.44 | 2.46 | 4.10 | 5.48 | 3.28 | 5.24 |
| CPlot (Liu et al., 2024a) | ViT-B | SBD | 1.34 | 1.48 | 1.40 | 2.18 | 4.05 | 5.29 | 3.05 | 4.95 |
| MFP (Lee et al., 2024) | ViT-B | SBD | 1.38 | 1.48 | 1.39 | 2.17 | 3.92 | 5.32 | 3.21 | 5.24 |
| GraCo (Zhao et al., 2024) | ViT-B | SBD | 1.34 | 1.46 | 1.33 | 2.07 | 4.36 | 5.49 | 3.22 | 4.65 |
| InterFormer (Huang et al., 2023) | ViT-B | C+L | - | 1.50 | - | 3.14 | - | 6.19 | 3.78 | 6.34 |
| EMC-Click (Du et al., 2023) | MiT-B0 | C+L | 1.86 | 2.02 | 1.80 | 2.80 | 5.74 | 7.95 | 4.32 | 7.09 |
| ClickAttention (Xu et al., 2024) | MiT-B0 | C+L | - | 1.66 | - | 2.65 | - | 5.44 | 3.73 | 5.92 |
| SegNext (Liu et al., 2024b) | ViT-B | C+L | 1.36 | 1.44 | 1.44 | 2.23 | 4.05 | 5.46 | 3.69 | 6.37 |
| MFP (Lee et al., 2024) | ViT-B | C+L | 1.34 | 1.42 | 1.35 | 1.90 | 3.37 | 4.81 | 3.26 | 5.34 |
| RefCut (Lin et al., 2025) | ViT-B | C+L | 1.34 | 1.46 | 1.39 | 1.89 | 3.65 | 4.99 | 3.21 | 5.36 |
| *Two-stage Local Refinement* | | | | | | | | | | |
| f-BRS-B (Sofiiuk et al., 2020) | ResNet-101 | SBD | 2.30 | 2.72 | - | 4.57 | 5.04 | 7.41 | 4.81 | 7.73 |
| FocusCut (Lin et al., 2022) | ResNet-101 | SBD | 1.46 | 1.64 | 1.81 | 3.01 | 4.85 | 6.22 | 3.40 | 5.31 |
| FocalClick (Chen et al., 2022) | HRNet-18s | SBD | 1.86 | 2.06 | - | 3.14 | 4.30 | 6.52 | 4.92 | 6.48 |
| FocalClick (Chen et al., 2022) | MiT-B0 | SBD | 1.66 | 1.90 | - | 3.14 | 4.34 | 6.51 | 5.02 | 7.06 |
| FocalClick (Chen et al., 2022) | HRNet-18s | C+L | 1.64 | 1.82 | - | 2.89 | 4.77 | 6.56 | 4.74 | 7.29 |
| FocalClick (Chen et al., 2022) | MiT-B0 | C+L | 1.40 | 1.66 | - | 2.27 | 4.04 | 5.49 | 4.56 | 6.86 |
| *One-stage Local Refinement* | | | | | | | | | | |
| FCFI (Wei et al., 2023) | ResNet-101 | SBD | 1.64 | 1.80 | 1.75 | 2.84 | 4.75 | 6.48 | 3.26 | 5.35 |
| EFPNet (ours) | ResNet-101 | SBD | 1.54 | 1.72 | 1.59 | 2.67 | 4.47 | 6.11 | 3.07 | 5.08 |
| EFPNet (ours) | ViT-B | SBD | 1.34 | 1.44 | 1.36 | 2.07 | 3.98 | 5.32 | 2.78 | 4.59 |
| FCFI (Wei et al., 2023) | HRNet-18s | C+L | 1.50 | 1.56 | 1.57 | 2.05 | 3.70 | 5.16 | 3.88 | 6.24 |
| EFPNet (ours) | HRNet-18s | C+L | 1.44 | 1.54 | 1.41 | 2.05 | 3.64 | 5.02 | 3.63 | 5.94 |
| EFPNet (ours) | ViT-B | C+L | 1.34 | 1.42 | 1.30 | 1.87 | 3.39 | 4.83 | 3.14 | 5.27 |

IoU at $x$-th click. Besides, we introduce a novel metric FIoU to measure the performance of FVP by calculating the average bounding box IoU between $\hat{v}$ and $v$ for all clicks.

**Implementation Details.** To fairly compare with FCFI, we adopt three fundamental network structures, including a) ResNet-101 (He et al., 2016) backbone with DeepLabV3+ decoder (Chen et al., 2018). b) HRNet-18s (Sun et al., 2019) backbone with OCR (Yuan et al., 2020) decoder. c) ViT-B (Dosovitskiy et al., 2020) backbone (pretrained by MAE (He et al., 2022)) with SimpleFPN (Liu et al., 2023) decoder. For each version, data augmentation is equally performed, such as random cropping, resizing, flipping and brightness/contrast. The learning rate is set to $5e^{-4}$ and attenuated by a cosine-annealing scheduler (Loshchilov & Hutter, 2016). We adopt Adam (Kingma & Ba, 2014) optimizer with a momentum of $\beta_1 = 0.9$ and $\beta_2 = 0.99$. Following (Sofiiuk et al., 2022; Liu et al., 2023), we train for 220 epochs on ResNet-101 and HRNet-18s, while for 55 epochs on ViT-B. All the experiments are conducted on Ubuntu-18.04 platform with 4 RTX 4090 GPUs. During the click simulation stage, we follow (Xu et al., 2016) to randomly preset positive and negative clicks, while the maximum number is set to 48. We utilize iterative training (Sofiiuk et al., 2022) up to 3 clicks.

### 4.2. Comparisons With State-Of-The-Arts

**Quantitative Analysis.** Firstly, We tabulate Number-of-Click (NoC) result in Table 1, where we categorize the mainstream methods based on whether one or more explicit refinement strategies are adopted. The *two-stage* means crop-and-refine the final prediction directly, while *one-stage* means crop-and-refine the previous feedback to indirectly guide the final prediction. Evidently, our method outperforms FCFI at around 0.2 to 0.4 NoC based on the same backbone (ResNet-101 and HRNet-18s) and the same training dataset (SBD and COCO+LVIS), while also outperforms all of the two-stage methods except for FocusCut results on GrabCut. This is mainly because FocusCut re-infers the whole network for several times, leading to huge computational burden. Moreover, when using ViT-B as backbone, our method also exceeds the latest counterparts based on vision transformer, *e.g*, EMC-Click, MFP and RefCut, which proves the efficacy of EFPNet.

Secondly, we measure the FIoU metrics within FocusCut, FocalClick, FCFI and EFPNet to assess the robustness of focus view. As shown in Table 2, our method manifests great potential into focus view perception when compared with others. It is obvious that FCFI obtains the worst FIoU score since they remain a solid scope of focus (see Figure 4).

*Table 2.* Comparison of other metrics on Berkeley and SBD datasets. The best results are marked as bold.

| Method | Berkeley | | | | | SBD | | | | |
|---|---|---|---|---|---|---|---|---|---|---|
| | 1-IoU | 5-IoU | 10-IoU | FIoU↑ | NoF%90↓ | 1-IoU | 5-IoU | 10-IoU | FIoU↑ | NoF%90↓ |
| CDNet$_{resnet101}$ (Chen et al., 2021) | 0.827 | 0.921 | 0.954 | - | 4 | 0.726 | 0.864 | 0.901 | - | 131 |
| FocusCut$_{resnet101}$ (Lin et al., 2022) | 0.809 | 0.933 | 0.951 | 0.607 | 3 | 0.704 | 0.889 | 0.926 | 0.493 | 147 |
| FocalClick$_{hrnet18s}$ (Chen et al., 2022) | 0.721 | 0.946 | 0.951 | 0.629 | 4 | 0.611 | 0.863 | 0.907 | 0.501 | 149 |
| FocalClick$_{mitb0}$ (Chen et al., 2022) | 0.749 | 0.957 | 0.962 | 0.654 | 1 | 0.665 | 0.894 | 0.924 | 0.519 | 120 |
| FCFI$_{resnet101}$ (Wei et al., 2023) | 0.800 | 0.943 | 0.956 | 0.117 | 3 | 0.753 | 0.901 | 0.923 | 0.094 | 112 |
| FCFI$_{hrnet18s}$ (Wei et al., 2023) | 0.818 | 0.958 | 0.965 | 0.114 | 0 | 0.697 | 0.895 | 0.921 | 0.106 | 108 |
| SegNext$_{vitb}$ (Liu et al., 2024b) | 0.841 | 0.958 | 0.967 | - | 2 | 0.733 | 0.897 | 0.922 | - | 117 |
| MFP$_{vitb}$ (Lee et al., 2024) | 0.855 | 0.962 | 0.965 | - | 1 | 0.769 | 0.906 | 0.928 | - | 101 |
| EFPNet$_{resnet101}$ (ours) | 0.837 | 0.945 | 0.956 | 0.609 | 2 | 0.782 | 0.904 | 0.923 | 0.500 | 109 |
| EFPNet$_{hrnet18s}$ (ours) | 0.851 | 0.964 | 0.966 | 0.641 | 1 | 0.724 | 0.901 | 0.923 | 0.514 | 104 |
| EFPNet$_{vitb}$ (ours) | **0.858** | **0.966** | **0.967** | **0.673** | **0** | **0.800** | **0.921** | **0.933** | **0.539** | **99** |

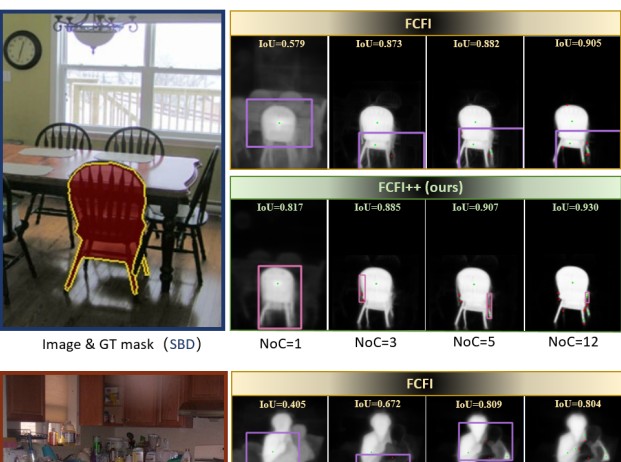

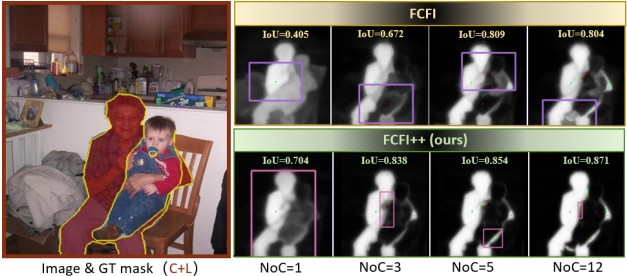

*Figure 4.* Qualitative comparison between FCFI and EFPNet using samples from SBD and COCO+LVIS datasets.

*Table 3.* Efficiency analysis under 3 different backbones.

| Backbone | Method | Params (M) | FLOPs (G) | SPC (s) |
|---|---|---|---|---|
| ResNet-101 | FocusCut (Lin et al., 2022) | 57.21 | 194.25 | 1.254 |
| | FocalClick (Chen et al., 2022) | 58.44 | 197.94 | 0.38 |
| | FCFI (Wei et al., 2023) | 58.66 | 195.72 | 0.023 |
| | EFPNet (ours) | 60.77 | 197.19 | 0.029 |
| HRNet-18s | RITM (Sofiiuk et al., 2022) | 4.22 | 11.21 | 0.032 |
| | FocalClick (Chen et al., 2022) | 4.37 | 11.78 | 0.026 |
| | FCFI (Wei et al., 2023) | 10.01 | 18.94 | 0.017 |
| | EFPNet (ours) | 10.83 | 20.47 | 0.020 |
| ViT-B | SimpleClick (Liu et al., 2023) | 96.64 | 99.69 | 0.039 |
| | SegNext (Liu et al., 2024b) | 102.45 | 154.36 | 0.044 |
| | MFP (Lee et al., 2024) | 97.27 | 105.45 | 0.037 |
| | EFPNet (ours) | 98.06 | 113.14 | 0.041 |

FCFI fails to focus on the salient object due to limited priors plus latent ambiguity in the clicked region, while our pipeline is capable of capturing precise user intentions given the least user prompt. We also visualize the concrete EFPNet workflow in Figure 5 to outline its variable states.

**Efficiency Analysis.** In Table 3, we print out time and capacity metrics for each model and its variants, relying on a single RTX 4090 card and $448 \times 448$ testing samples. Our EFPNet only introduces a slight model parameters and inference cost (mainly from the FVP and CFI module) compared with FCFI but a salient performance gain. When compared using ViT-B, our time cost is also competitive, with only 32ms backward from SimpleClick and 2ms from MFP.

### 4.3. Ablation Studies

We perform a comprehensive analysis w.r.t three core components in EFPNet, including focus view predictor (FVP), triple-affinity correction (TAC), and clicked-with-focus integration (CFI). We start from a plain FCFI baseline where we adopt ViT-B as backbone and train on COCO+LVIS. Next, we progressively add these components to replace the FCFI counterpart and record the NoC score. Ablation results on Berkeley and DAVIS datasets are shown in Table 4.

**Analysis for FVP.** When only FVP is utilized, there's a huge improvement on SBD dataset (0.17 for NoC%85 and

Moreover, even if FocusCut and FocalClick could produce adaptive focus view, the accuracy will be restricted when current prediction has no difference with previous feedback, especially when the quality of latter is low.

We further provide $x$-IoU metric for convergence analysis, where $x$ is set to 1, 5, 10. When under the same backbone, EFPNet overwhelms especially in 1-IoU, illustrating that the design of "switching to whole instance" for FVP in one-click scenario manifests the global perception, which is beneficial for *feedback integrity* in the former interaction.

**Qualitative Analysis.** In Figure 4, we conduct visualizations w.r.t the interacting process of FCFI and EFPNet. Clearly, the convergence speed of EFPNet is significantly faster than FCFI especially when NoC=1. We argue that

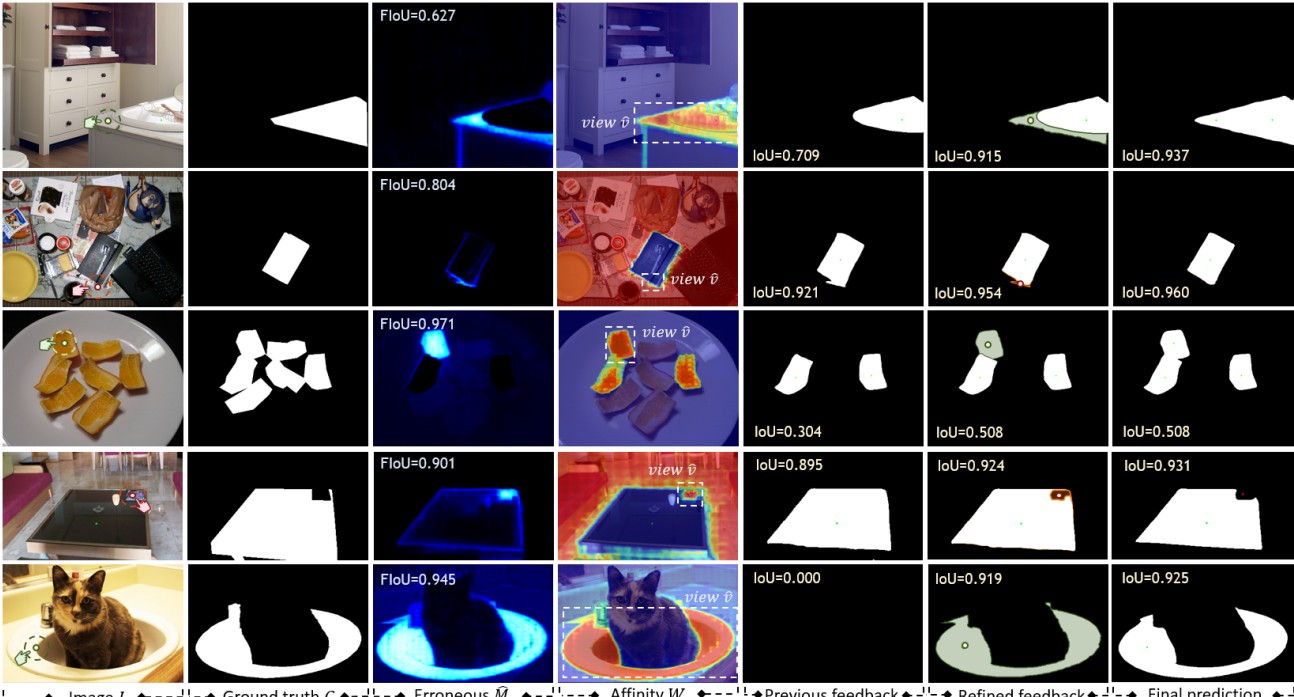

*Figure 5.* Visualization for the whole pipeline of EFPNet, where core intermediate variables are displayed.

0.28 for NoC%90) and Berkeley dataset (0.09 for NoC%85 and 0.08 for NoC%90), which proves the significance of FVP. This is because in contrast with solid focus view which brings about redundancies or detail leakage, our method could dynamic focus view given arbitrary pairs of click position and previous feedback.

**Analysis for TAC**. The choice of affinity function in feedback correction process is of great necessity, where our proposed TAC considers image (low-level), feature (mid-level) and mask (high-level) collectively, each embedded with unique prior that others may not contain, which boosts the correction accuracy (0.1 NoC%85 and 0.12 NoC%90 increase for SBD dataset). Moreover, we further explore the impact of affinity weighting terms $\lambda_P$, $\lambda_F$ and $\lambda_M$. In Table 5, we first record the results separately for each type of affinity by setting other two to zero, then an average mixture of three (0.5, 0.5, 0.5) is evaluated. Despite manually setting, we also consider them as learnable parameters, where we find it reaches the climax effect among all conditions—up to 0.29 NoC%90 performance gain.

**Analysis for CFI**. CFI consists of two submodules—inter-focus contrastive learning (FCL) and intra-focus attention (FA), which dedicates both 0.12 NoC%85 and NOC%90 increase for SBD dataset. For FCL, it provides strict inter-focused affinity rules for click and temporal consistency, acting as a reliance for FA to capture the pixel-region correlation within each focus window, which ensures high cohesion and high utilization rate of focus prior.

*Table 4.* A plug-in analysis for each core component in EFPNet.

| Component | | | | Berkeley | | SBD | |
| --- | --- | --- | --- | --- | --- | --- | --- |
| FVP | TAC | FCL | FA | NoC%85 | NoC%90 | NoC%85 | NoC%90 |
| | | | | 1.49 | 1.97 | 3.49 | 5.83 |
| ✓ | | | | 1.40 | 1.91 | 3.32 | 5.55 |
| | ✓ | | | 1.45 | 1.94 | 3.39 | 5.71 |
| ✓ | ✓ | | | 1.34 | 1.89 | 3.26 | 5.39 |
| ✓ | ✓ | ✓ | | 1.35 | 1.88 | 3.18 | 5.41 |
| ✓ | ✓ | | ✓ | 1.33 | 1.87 | 3.22 | 5.36 |
| ✓ | ✓ | ✓ | ✓ | **1.30** | **1.87** | **3.14** | **5.27** |

*Table 5.* Comparison with different TAC's affinity weighting term.

| TAC's Weighting | | | Berkeley | | SBD | |
| --- | --- | --- | --- | --- | --- | --- |
| $\lambda_P$ | $\lambda_F$ | $\lambda_M$ | NoC%85 | NoC%90 | NoC%85 | NoC%90 |
| 1.0 | 0.0 | 0.0 | 1.51 | 2.08 | 3.43 | 5.69 |
| 0.0 | 1.0 | 0.0 | 1.45 | 1.94 | 3.35 | 5.56 |
| 0.0 | 0.0 | 1.0 | 1.39 | 1.94 | 3.38 | 5.44 |
| 0.5 | 0.5 | 0.5 | 1.36 | 1.93 | 3.22 | 5.31 |
| *Learnable* | | | **1.30** | **1.87** | **3.14** | **5.27** |

## 5. Conclusion

We propose EFPNet, a FCFI-modified pipeline that aims to leverage focus prior in a robust and efficient paradigm by improving its core components, including focus view predictor, feedback correction based on novel triple-affinity principle, and powerful feedback integration scheme. Extensive experiments have demonstrated its feasibility in various scenario and importance of focus-oriented modeling.

## Impact Statement

This paper presents work whose goal is to advance the field of Machine Learning. There are many potential societal consequences of our work, none which we feel must be specifically highlighted here.

## Acknowledgments

This work was supported in part by the National Natural Science Foundation of China under Grant 62372235, Grant 62406069, in part by the China Postdoctoral Science Foundation, under Grant 2024M750425.

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

## A. Notations

A list of symbols and their illustrations is shown in Table 6, including variables, hyperparameters, and functions.

## B. Additional Training Details

**Loss Function**. The training objective in our framework is comprised of error prediction $\hat{E}_m$, refined previous feedback $M_{t-1}^r$, final prediction $M_t$, and focused feature $F^l$. The former three terms are supervised by normalized focal loss (NFL), while the latter adopts the inter-focus contrastive loss $L_{fcl}$ for click-based pairwise optimization. The full loss is written as follows:

$$L = \omega_1 L_{nfl}(\hat{E}_m, E_m) + \omega_2 L_{nfl}(M_{t-1}^r, G)$$
$$+ \omega_3 L_{nfl}(M_t, G) + \omega_4 L_{fcl}(F^l, \mathcal{C}, \{\hat{v}_i\}_{i=1}^t) \quad (17)$$

where $\omega_1$, $\omega_2$, $\omega_3$ and $\omega_4$ are set to 0.5, 0.5, 1.0 and 0.5, respectively.

**Ambiguity for Global View**. When interaction starts, user is inclined to set a primitive click around the structural or mass center of an latent object, which is coordinated with "global view" perception. However, a scene may contain objects with complicated hierarchies and spatial/semantic relations, which brings about confusion to comprehending the precise user intention from several candidates regions. For example, one could not distinguish from (a) A hat. (b) A man with a hat. (c) A man with a hat is sitting on a chair—If a user's first click is located on the "hat". To handle this ambiguous situation, we construct a hierarchical tree using LVIS dataset, with root nodes as a complete object and leaf nodes as its candidate subpart regions. During training, if first click is sampled nearby any subpart region but far away from complete object center, we regulate the global view to perceive the subpart region; If the center of object and its subpart (maybe more than one) overlaps, we select the subpart with *minimal area* as global view candidate. This design dramatically reduce the risk of ambiguous network response during the former period of interaction.

**Pseudo View for Preset Clicks**. It has been proved by previous methods that directly training from one click will lead to poor optimization, since the network requires enriched interaction priors to construct a precise mapping from clicks to instance mask, which is difficult to learn when the click is sparse. Therefore, a multi-clicks sampling strategy is adopted to initially set a collection of preset-clicks on the mask during data-fetching process. However, the above training process is not aligned with real-world user behavior, since click is progressively added from 1 to $N$, where predicted focus view $\{\hat{v}_i\}_{i=1}^N$ for historical clicks is unavailable for preset-clicks due to lack of temporal information, which is indispensable for the computation of our clicked-with-focus integration (CFI) module.

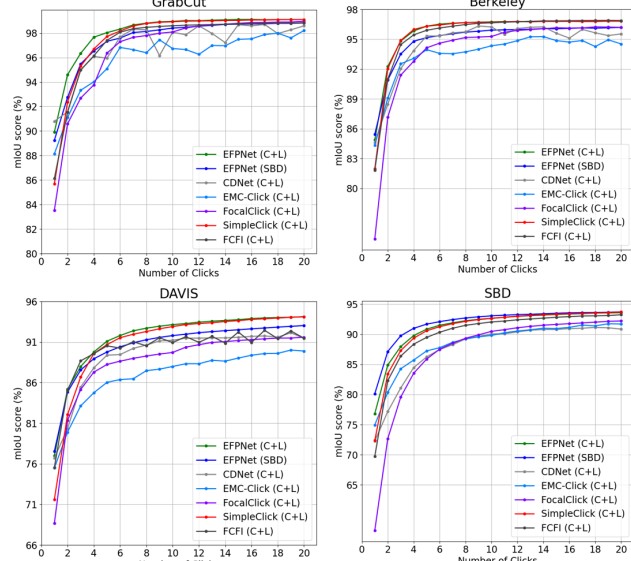

*Figure 6.* Curves to measure the change of IoU with the number of clicks on four datasets.

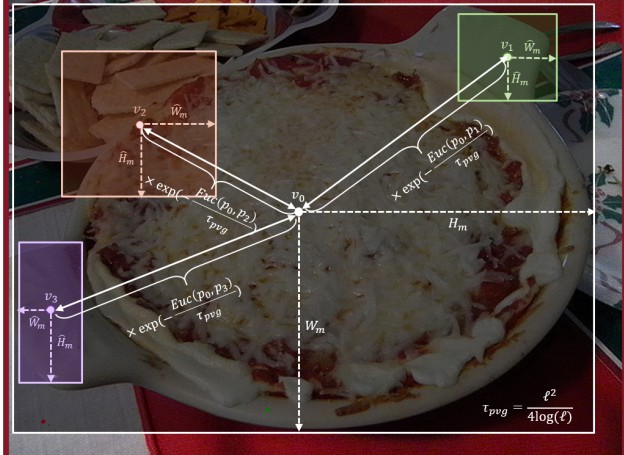

*Figure 7.* The pseudo view generation process for training.

Hence, we propose a *pseudo view generation* strategy for each preset click, as shown in Figure 7. Our main idea is to model each view as a relative decay of global view (range of object mask) in terms of horizontal and vertical distance between click and object mask center $c_0$. Firstly, we calculate the tightest bounding box with shape $\ell \in \{H_m, W_m\}$ for ground-truth $G$, then for $i$-th click $c_i$ with **positive property**, the shape $\hat{\ell} \in \{\hat{H}_m, \hat{W}_m\}$ of its pseudo view is formulated as follows:

$$\hat{\ell} = \ell \times \exp\left(-\frac{Euc(c_0, c_i)}{\tau_{pvg}}\right) \quad (18)$$

$$\tau_{pvg} = \frac{\ell^2}{4\log(\ell)} \quad (19)$$

where $Euc(\cdot, \cdot)$ is euclidean distance function, $\tau_{pvg}$ is a

*Table 6.* Notations for the paper, where variables, hyperparameters and functions are abbreviated as $\mathcal{V}$, $\mathcal{H}$ and $\mathcal{F}$.

| Symbol | Type | Value | Description | Symbol | Type | Value | Description |
|--------|------|-------|-------------|--------|------|-------|-------------|
| $I$ | $\mathcal{V}$ | $\mathbb{R}^{H \times W \times 3}$ | *Input image* | $F_d$ | $\mathcal{V}$ | $\mathbb{R}^{\frac{H}{4} \times \frac{W}{4} \times d}$ | *Mask decoder feature* |
| $\mathcal{C}$ | $\mathcal{V}$ | $\{(u_i, v_i, p_i)\}_{i=1}^N$ | *User clicks* | $\mathbf{z}$ | $\mathcal{V}$ | $\mathbb{R}^d$ | *Learnable token* |
| $c_0$ | $\mathcal{V}$ | $(u_0, v_0, p_0)$ | *First click* | $\mathbf{z}^*$ | $\mathcal{V}$ | $\mathbb{R}^{(t+1) \times d}$ | *Learnable token with click embedding* |
| $c_{new}$ | $\mathcal{V}$ | $(u_t, v_t, p_t)$ | *New click (in t-th round)* | $\tau$ | $\mathcal{H}$ | $\mathbb{R}$ | *Threshold for FVP* |
| $S$ | $\mathcal{V}$ | $\mathbb{R}^{H \times W \times 2}$ | *Disk map for all clicks $\mathcal{C}$* | $\sigma_p$ | $\mathcal{H}$ | $\mathbb{R}$ | *Decay coefficient for pixel affinity* |
| $S_{new}$ | $\mathcal{V}$ | $\mathbb{R}^{H \times W \times 2}$ | *Disk map for new click $c_{new}$* | $\gamma$ | $\mathcal{H}$ | $\mathbb{R}$ | *Decay coefficient for mask affinity* |
| $E_p, E_n$ | $\mathcal{V}$ | $[0,1]^{H \times W}$ | *Erroneous mask between $M_{t-1}$ and $G$* | $\lambda_P, \lambda_F, \lambda_M$ | $\mathcal{H}$ | $\mathbb{R}$ | *Affinity weighting term* |
| $O_p, O_n$ | $\mathcal{V}$ | $\mathbb{N}^+$ | *Pixel summation for $E_p$ and $E_n$* | $\tau_p, \tau_n$ | $\mathcal{H}$ | $\mathbb{R}$ | *Threshold for FCL* |
| $E_m, \hat{E}_m$ | $\mathcal{V}$ | $[0,1]^{H \times W}$ | *Click-centered max-connected region of $E_p$ or $E_n$* | $\phi_d(\cdot)$ | $\mathcal{F}$ | $-$ | *Classifier for FVP* |
| $v_i, \hat{v}_i$ | $\mathcal{V}$ | $\mathbb{R}^4$ | *GT/predicted focus view for i-th click* | $\phi_p(\cdot)$ | $\mathcal{F}$ | $-$ | *Patch embedding layer* |
| $G$ | $\mathcal{V}$ | $\{0,1\}^{H \times W}$ | *GT mask* | $BBox(\cdot)$ | $\mathcal{F}$ | $-$ | *BBox inference with $\hat{E}_m$* |
| $M_{t-1}$ | $\mathcal{V}$ | $[0,1]^{H \times W}$ | *Previous feedback* | $W_P(\cdot, \cdot)$ | $\mathcal{F}$ | $-$ | *Pixel affinity* |
| $M_{t-1}^r$ | $\mathcal{V}$ | $[0,1]^{H \times W}$ | *Refined feedback* | $W_F(\cdot, \cdot)$ | $\mathcal{F}$ | $-$ | *Feature affinity* |
| $M_{v_i}$ | $\mathcal{V}$ | $[0,1]^{H \times W}$ | *Indicator mask for i-th click* | $W_M(\cdot, \cdot)$ | $\mathcal{F}$ | $-$ | *Mask affinity* |
| $M_t$ | $\mathcal{V}$ | $[0,1]^{H \times W}$ | *Final prediction in t-th round* | $\kappa_p(\cdot, \cdot)$ | $\mathcal{F}$ | $-$ | *Distance function for pixel affinity* |
| $K_p, K_n$ | $\mathcal{V}$ | $\mathbb{N}^+$ | *Number of sampled positive/negative click pairs* | $\kappa_m(\cdot, \cdot)$ | $\mathcal{F}$ | $-$ | *Distance function for mask affinity* |
| $F$ | $\mathcal{V}$ | $\mathbb{R}^{\frac{H}{16} \times \frac{W}{16} \times d}$ | *Backbone feature* | $L_{fcl}(\cdot, \cdot, \cdot)$ | $\mathcal{F}$ | $-$ | *Inter-focus contrastive loss* |
| $F^l$ | $\mathcal{V}$ | $\mathbb{R}^{t \times d}$ | *Focused feature* | $\mathbb{I}[\cdot]$ | $\mathcal{F}$ | $-$ | *Indicator function* |
| $F_c$ | $\mathcal{V}$ | $\mathbb{R}^{t \times d}$ | *Click embedding* | $S(\cdot, \cdot)$ | $\mathcal{F}$ | $-$ | *Cosine-based similarity* |

*Table 7.* Comparion with 9 SAM-style methods on HQSeg44K and DAVIS datasets. The testing resolution is scaled from $1024^2$ to $2048^2$.

| Model | Backbone | Train set | Resolution | Inference time SPC (ms) | HQSeg44K 5-mIoU | NoC%90 | NoC%95 | DAVIS 5-mIoU | NoC%90 | NoC%95 |
|-------|----------|-----------|------------|------|------|------|------|------|------|------|
| SAM (Kirillov et al., 2023) | ViT-B | SA-1B | $1024^2$ | 15 | 86.16 | 7.46 | 12.42 | 90.95 | 5.14 | 10.74 |
| SAM (Kirillov et al., 2023) | ViT-H | SA-1B | $1024^2$ | 32 | 87.21 | 6.85 | 11.57 | 90.82 | 5.20 | 10.04 |
| MobileSAM (Zhang et al., 2023) | ViT-T | SA-1B | $1024^2$ | 16 | 81.98 | 8.70 | 13.83 | 89.18 | 5.83 | 12.74 |
| EfficientSAM (Xiong et al., 2024) | ViT-T | SA-1B+ImageNet | $1024^2$ | 13 | 77.9 | 10.11 | 14.60 | 85.26 | 7.37 | 14.28 |
| EfficientSAM (Xiong et al., 2024) | ViT-S | SA-1B+ImageNet | $1024^2$ | 15 | 79.01 | 8.84 | 13.18 | 87.55 | 6.37 | 12.26 |
| HQSAM (Ke et al., 2023) | ViT-B | SA-1B+HQ | $1024^2$ | 16 | 89.85 | 6.49 | 10.79 | 91.77 | 5.26 | 10.74 |
| SegNext (SA * 1) (Liu et al., 2024b) | ViT-B | C+L | $1024^2$ | 53 | 85.41 | 7.47 | 11.94 | 90.13 | 5.46 | 13.31 |
| SegNext (SA * 2) (Liu et al., 2024b) | ViT-B | C+L+HQ | $1024^2$ | 68 | 91.75 | 5.32 | 9.42 | 91.87 | 4.43 | 10.73 |
| HRSAM++ (Huang et al., 2024a) | ViT-B | C+L+HQ | $1024^2$ | 24 | 90.94 | 5.87 | 9.89 | 91.06 | 4.92 | 11.93 |
| Inter2Former (Huang et al., 2025) | ViT-B | C+L+HQ | $1024^2$ | 37 | 91.48 | 5.36 | 9.29 | 90.82 | 4.9 | 11.33 |
| EFPNet (ours) | ViT-B | C+L+HQ | $1024^2$ | 46 | 91.87 | 5.32 | 9.39 | 92.04 | 4.36 | 10.65 |
| HRSAM++ (Huang et al., 2024a) | ViT-B | C+L+HQ | $2048^2$ | 33 | 92.61 | 4.87 | 8.38 | 92.11 | 4.57 | 9.59 |
| Inter2Former (Huang et al., 2025) | ViT-B | C+L+HQ | $2048^2$ | 46 | 92.28 | 4.58 | 7.79 | 91.30 | 4.33 | 8.45 |
| EFPNet (ours) | ViT-B | C+L+HQ | $2048^2$ | 55 | 93.02 | 4.52 | 7.78 | 92.09 | 4.29 | 8.48 |

normalizing term which regulates focus view $v_i$ to cover only 1 pixel if click $c_i$ is located right on the boarder of $G$.

For click with **negative property**, we follow the same strategy in FocusCut to introduce a set of control parameter $\alpha$ and $\beta$, which formulated as follows:

$$\hat{H}_m = \alpha \cdot H, \hat{W}_m = \beta \cdot W \qquad (20)$$

where $H$, $W$ is the size of image $I$. In practice, $\alpha$ and $\beta$ are randomly sampled in zone $[0.2, 0.8]$.

## C. Additional Results

**IoU Curve**. In Figure 6, we record the change of prediction metric (IoU) *w.r.t.* interaction rounds on GrabCut, Berkeley, DAVIS and SBD dataset, where our EFPNet is plotted as green line (trained on COCO+LVIS) and indigo line (trained on SBD). The comparison between EFPNet and other baselines shows a significant gain on both convergence speed and 1-click accuracy, due to our modified use of focus view in contrast with counterparts methods using similar con-

cept—ranging from prediction, correction and integration, where the prior of focus takes part in almost every stage of regional refinement.

**Evaluation with SAM**. A variety of SAM-style research works (Kirillov et al., 2023; Xiong et al., 2024; Liu et al., 2024b; Huang et al., 2025) has been prevailing in recent years, which injects zero-shot and multi-modal capability into classical interactive segmentation. As a novel benchmark, HQSeg44K dataset is introduced in HQSAM (Ke et al., 2023) for both training and testing purposes, which consists of 44k fine-grained annotated masks with more than 1000 categories. Based on our raw EFPNet version (trained on COCO+LVIS using ViT-B backbone), we setup an extra fine-tuning stage on HQSeg44K training set for 1 epoch, following (Ke et al., 2023; Huang et al., 2025) protocols. We complement with a detailed comparison among up to 9 SAM-style methods on HQSeg44K and DAVIS dataset (scaling from $1024^2$ to $2048^2$ resolution), where we record *second-per-click* (SPC) on 1 RTX 4090 GPU, 5-mIoU and *number-of-click* (NoC) metrics in Table 7. Apparently, our

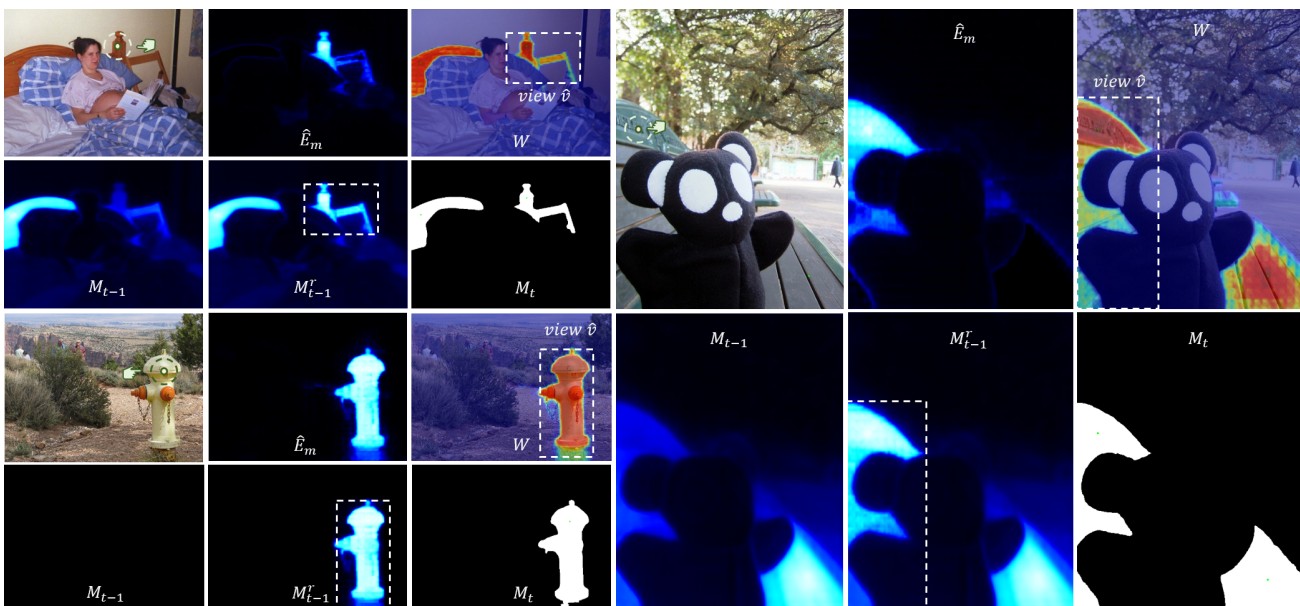

*Figure 8.* Additional visualization of EFPNet pipeline, where we illustrate the core variable state during inference.

*Table 8.* Comparison across different datasets with varying $\tau$ for FVP.

| $\tau$ | Berkeley | | | DAVIS | | | SBD | | | HQSeg44K | | |
|---|---|---|---|---|---|---|---|---|---|---|---|---|
| | NoC%90 | FIoU | 1-mIoU | NoC%90 | FIoU | 1-mIoU | NoC%90 | FIoU | 1-mIoU | NoC%90 | FIoU | 1-mIoU |
| 0.30 | 1.90 | 0.652 | 0.854 | 4.86 | 0.607 | 0.773 | 5.29 | 0.524 | 0.761 | 4.54 | 0.635 | 0.816 |
| 0.40 | 1.88 | 0.664 | 0.854 | 4.85 | 0.622 | 0.775 | 5.27 | 0.531 | 0.763 | 4.51 | 0.638 | 0.823 |
| 0.45 | **1.87** | 0.671 | 0.856 | **4.83** | 0.630 | 0.776 | **5.26** | 0.536 | **0.766** | **4.51** | 0.640 | **0.823** |
| 0.50 | **1.87** | **0.673** | **0.858** | **4.83** | **0.635** | **0.776** | 5.27 | **0.539** | 0.764 | 4.52 | **0.641** | 0.822 |
| 0.55 | **1.87** | **0.673** | 0.856 | 4.85 | 0.634 | 0.775 | 5.27 | 0.538 | 0.764 | 4.53 | 0.639 | 0.822 |
| 0.60 | 1.89 | 0.669 | 0.852 | 4.86 | 0.629 | 0.773 | 5.29 | 0.533 | 0.762 | 4.55 | 0.636 | 0.819 |
| 0.70 | 1.89 | 0.661 | 0.848 | 4.89 | 0.618 | 0.769 | 5.33 | 0.526 | 0.754 | 4.61 | 0.631 | 0.813 |

EFPNet surpasses all mainstream methods (despite a few exceptions) while maintaining an acceptable computation cost, this is because when testing resolution is uprising, it is prone for some lightweight SAMs (Zhang et al., 2023; Xiong et al., 2024) to fall into detail collapse, since it requires plenty of computational blocks to handle fairly-large global context. Even methods with the latest elaborate designs (*i.e.*, flash-attention (Huang et al., 2024a), Mixture-of-Expert (Huang et al., 2025)) also suffers from slow convergence speed (more iteration steps to reach satisfaction), which boosts the significance of coarse-to-fine local refinement instead of "inference as a whole" for each interaction round.

**Ablation for FVP**. In FVP, we use $\tau$ as the binary threshold for focus constraint, which determines the elasticity of focus and is manually controllable. To achieve a fair comparison, a default value $\tau = 0.5$ is applied in both training and inference for all experiment records in the main paper, without any model or datasets-specific tuning. Meanwhile, we're also interested in searching for the best $\tau$ for different datasets, where we perform a $\tau$-sweep evaluation ranging

from 0.3 to 0.7 using four datasets. Our baseline model is trained with ViT-B backbone on COCO+LVIS. As is shown in Table 8, all metrics—whatever NoC/1-mIoU/FIoU, present a U-style or invertible U-style distribution around a respective $\tau$ values for each dataset, which equals or slightly vary from 0.5 due to different object statistics. For Berkeley/DAVIS filled with salient and mostly single-count object, their NoC metrics remain constant where $\tau \in [0.45, 0.55]$; while for SBD/HQSeg44K filled with multi-count texture-detailed objects, their optimal $\tau$ becomes 0.45 and 0.4. We argue that the lack of focus details ($\tau$ is large) is much more fatal than content redundancy ($\tau$ is small), especially when it's difficult to precisely handle the erroneous region (See FIoU metric) for complex samples. It's also easy to observe the abrupt decline of 1-IoU metric when $\tau$ is lifted in zone $[0.5, 0.7]$, compared to flat change in $[0.3, 0.5]$. Meanwhile, we must be cautious not to set an extremely lower $\tau$ since the redundancy will overbalance which causes unstable output.

**More Visualization**. Please refer to Figure 8 for visualization of EFPNet's core variables.

