# OpenReview forum: "Interactive Segmentation with Elaborate Focus Prior"
_ICML.cc/2026/Conference — ICML 2026 regular_

### Official Review · Reviewer_GnCE · 2026-02-26

**Soundness:** 3
**Presentation:** 3
**Significance:** 3
**Originality:** 3
**Overall Recommendation:** 4
**Confidence:** 3

**Summary:**

This paper proposes EFPNet for click-based interactive segmentation, aiming to improve local correction efficiency and accuracy by replacing fixed or two-stage “focus view” refinement with a learned dynamic focus prior. The method includes (1) a Focus View Predictor that estimates an error region around the latest click to define a focus box, (2) a Triple-Affinity Correction module combining pixel-, feature-, and mask-affinity to update the previous mask inside the focus, and (3) a click-aware integration module using inter-focus contrastive learning and intra-focus attention to better retain and utilize historical clicks. Experiments across multiple backbones and standard benchmarks report reduced clicks to reach target IoU and improved focus quality, with modest runtime overhead versus fixed-focus baselines.

**Compliance With Llm Reviewing Policy:**

Affirmed.

**Final Justification:**

The author has addressed my concerns.

**Key Questions For Authors:**

1. Could you provide qualitative failure cases where the focus predictor mislocalizes (e.g., thin structures, ambiguous boundaries, cluttered scenes), and explain how the downstream correction behaves in those cases?
2. How large is the train–test gap introduced by pseudo focus views for preset clicks, and can you quantify its impact by training with true sequential interactions only (even on a subset) for comparison?

**Limitations:**

yes

**Strengths And Weaknesses:**

Strengths
1. Clear motivation: dynamic, click-driven local refinement for interactive correction.
2. Strong empirical coverage across datasets and multiple backbones/decoders.

Weaknesses
1. The focus box is derived by thresholding the predicted error map, and the method may be sensitive to the choice of threshold and post-processing details; this sensitivity is not thoroughly characterized.
2. The training-time handling of preset clicks via pseudo focus views introduces a train–test mismatch, and the impact of this approximation on the attention/integration module is not fully isolated.

---

> ### Author Rebuttal · Authors · 2026-03-31
>
> Q1: Qualitative failure cases.
>
> A1: From our qualitative analysis, we observe that the FVP occasionally struggles with: (1) extremely thin structures (< 3 pixels wide) where the error mask prediction is noisy, (2) highly cluttered scenes with multiple overlapping objects, and (3) objects with very low contrast against the background. These failure modes are common across interactive segmentation methods and represent fundamental challenges in the field. As discussed with **Reviewer Uxsy**, the most robust way to handle these situation is to perform click-aware max-connected error correction instead of using full error masks.
>
> Q2: Thresholding in FVP.
>
> A2: We've conducted a series of extra comparison studies. Please refer to the rebuttal in **Reviewer Uxsy**~
>
> Q3: train–test gap.
>
> A3: We've also done this comparison and analysis. Please refer to the first table in the rebuttal of **Reviewer msMH**~

---

> > ### Author Rebuttal · Reviewer_GnCE · 2026-04-03
> >
> > Thank you for your response.

---

> > > ### Author Response · Authors · 2026-04-07
> > >
> > > Thank you for your approval to this paper！

---

### Official Review · Reviewer_msMH · 2026-03-03

**Soundness:** 2
**Presentation:** 3
**Significance:** 3
**Originality:** 2
**Overall Recommendation:** 4
**Confidence:** 2

**Summary:**

This paper introduces EFPNet, a novel one-stage framework for interactive image segmentation that enhances regional refinement by employing an "Elaborate Focus Prior." The key insight is to move beyond using a fixed, static region around user clicks for refinement, as done in the prior state-of-the-art method FCFI. EFPNet's main contributions are threefold: 1) A Focus View Predictor (FVP) that dynamically predicts an adaptive focus region by estimating an error mask from the previous interaction, allowing refinement to concentrate precisely where needed. 2) A Triple-Affinity Correction (TAC) module that refines the historical segmentation mask within the focus region using a combination of pixel-, feature-, and mask-level affinities, leading to more robust corrections. 3) A Clicked-with-Focus Integration (CFI) module that stabilizes and enhances the integration of corrected feedback through inter-focus contrastive learning and intra-focus attention mechanisms. Extensive experiments on standard benchmarks demonstrate that EFPNet achieves superior performance in reducing the number of required user clicks while maintaining efficient inference, effectively addressing the limitations of both two-stage and previous one-stage refinement methods.

**Compliance With Llm Reviewing Policy:**

Affirmed.

**Final Justification:**

Thank you for your response. Your response resolved all my questions.

**Key Questions For Authors:**

Q1: The description of the FVP's behavior for the first click (t=1) is ambiguous. During training, the FVP is supervised using the ground truth G as the error mask E_m (since M_{t-1} is all zeros). Could you clarify the precise inference mechanism and input for the FVP in this scenario during testing, where G is unavailable?

Q2: The pseudo view generation strategy for preset clicks (Appendix B, Eqs. 18-19) uses a specific exponential decay formula. What is the theoretical or empirical justification for this particular form? Was it validated against alternative strategies (e.g., based on image features or other training schemes)?

Q3: The FIoU metric demonstrates your FVP's ability to predict the focus region. However, the ablation study (Table 4) does not isolate how the quality of this prediction directly drives the final NoC improvement. Could you provide an analysis (e.g., a correlation study between per-sample FIoU and the reduction in clicks needed) to more conclusively argue that the performance gain stems precisely from better adaptive focus, rather than just from increased model capacity?

Q4: The paper positions itself as an improved version of FCFI. To highlight a higher level of conceptual novelty, could you further articulate the fundamental difference between your learnable, error-mask-based focus prediction (FVP) and the difference-based focus estimation in two-stage methods (e.g., FocalClick)? Beyond engineering efficiency (one-stage vs. two-stage), what new understanding or capability does the FVP enable regarding modeling user intent or error correction that was not possible with prior paradigms?

If the above question can be resolved, I am willing to increase my score.

**Limitations:**

Performance boundaries: Are there specific object types (e.g., highly transparent, filamentous, or heavily occluded objects) or scene conditions where the adaptive focus predictor (FVP) or the triple-affinity correction (TAC) might fail or underperform?

Generalization scope: The modules are designed for the click-based IS pipeline. How might this design limit the method's applicability to other interactive paradigms (e.g., scribbles, boxes) or its ability to be integrated into broader segmentation frameworks?

Efficiency trade-off: The increase in inference time (SPC) should be honestly framed as a limitation to be weighed against the accuracy gain, especially for real-time applications.

**Strengths And Weaknesses:**

Strengths

The paper's primary strength lies in its technical soundness and comprehensive experimental validation. The authors present a well-motivated, systematic enhancement of the FCFI framework through three novel and coherently designed components (FVP, TAC, CFI). The claims of superior performance are convincingly supported by extensive benchmarking across multiple datasets, backbones, and training sets, following established protocols. The work is clearly presented, reproducible, and offers a measurable advancement in efficiency for the important task of interactive segmentation.

Weaknesses

1. Methodological Gaps and Insufficient Analysis. Certain technical details lack rigor, and the experimental analysis could be more profound. Specifically:
(a) The behavior of the Focus View Predictor (FVP) during the first click is ambiguously described, creating a disconnect between its training (which uses ground truth) and inference logic.
(b) The "pseudo view" generation strategy for training is based on an unsubstantiated heuristic.
(c) The ablation studies do not deeply investigate the critical link between the quality of the predicted adaptive focus (FIoU) and the final performance gain.
(d) The discussion underplays the non-trivial increase in inference time (~17-26%) for a task prioritizing real-time interaction.

2. Constrained Originality and Incremental Nature. The core contribution is an expert-level redesign of the existing FCFI pipeline. While the integration of ideas (adaptive focus prediction, multi-modal affinity, historical focus attention) is creative and effective, the work is fundamentally anchored in improving a specific prior method. It demonstrates solid engineering but does not propose a new paradigm or offer a fundamental conceptual shift regarding "focus" in interactive segmentation.

3. Specialized Impact and Scope. The performance gains, though consistent, are incremental. The proposed modules are highly specialized and tightly coupled to the click-based refinement sub-task within a specific pipeline. This limits the work's direct applicability to broader problems in segmentation or interactive learning, and thus its potential for inspiring wide-ranging future research directions.

---

> ### Author Rebuttal · Authors · 2026-03-31
>
> Q1: FVP's behavior for the first click.
>
> A1: Whatever at first or subsequent clicks, we both need ground-truth $G$ to supervise FVP during training. Do you mean that we just directly set $G$ as the output of FVP instead of forwarding FVP when training on the first-click? It doesn't make that sense, since FVP is end-to-end. When $t=1$, FVP receives image, click, and zero-mask prior feedback $M_0$ as input, then outputs a complete object foreground mask for focus extraction, which plays the role of coarse **logit production**. The above process remains consistent for both training and testing period. The key insight is that the FVP architecture is designed to handle both scenarios: when $M_{t-1}$ is all-zero (first click), it performs instance segmentation; when $M_{t-1}$ contains prior feedback (subsequent clicks), it performs error prediction. This unified design eliminates the need for separate first-click handling logic.
>
> Q2: Pseudo-view strategy.
>
> A2: From a macro perspective, we strive to follow the realistic user interaction patterns during the training period--starting with broad intentions (object center) and progressively narrow their focus (object borders). From a micro perspective, the reason we adopt exponential decay formula mainly refers to existing empirical design and we perform extensive comparison with some candidate options--i.e., *linear decay*, *constant* (0.3 as FCFI) and *w/o* (always train from the first round without click preset), where we record their result in the following table (ViT-B backbone trained with COCOLVIS). Evidently, *exponential* slightly surpasses *linear* with 0.04/0.05/0.02 NoC%90 on Berkeley/DAVIS/SBD while moderately surpasses *constant* with 0.08/0.16/0.11 NoC%90, which indicates the efficacy of our design. Note that we don't desire an absolute real-and-precise focus view for historical clicks since they're only in charge of accumulating features for Clicked-with-Focus Integration (CFI) where proper deviation is allowed.
> | Pseudo Formula | Berkeley NoC@90 | Berkeley 1-IoU | DAVIS NoC@90 | DAVIS 1-IoU | SBD NoC@90 | SBD 1-IoU |
> |:--------------|----------------:|---------------:|-------------:|------------:|-----------:|----------:|
> | w/o           | 2.19            | 0.835          | 5.24         | 0.751       | 5.65       | 0.743     |
> | constant   | 1.95            | 0.847          | 4.99         | 0.767       | 5.38       | 0.755     |
> | linear        | 1.91            | 0.852          | 4.88         | 0.773       | 5.29       | 0.762     |
> | **exponential** | **1.87**      | **0.858**      | **4.83**     | **0.776**   | **5.27**   | **0.764** |
>
> Q3: Correlation between FIoU and NoC.
>
> A3: Besides our ablation study, Figure 4 also manifests how the IoU changes w.r.t solid and adaptive focus view on the same sample. The broad increase of FIoU is meant to lift up the interaction convergence speed, which reduces NoC. In our final revision, we'll provide more qualitative figure to clarify the relation of FIoU, IoU and NoC from click 1st to 20th. Meanwhile, we prefer to quantitatively analyze how FIoU affects NoC by adding an extra comparison: 1. Baseline EFPNet (with FVP prediction). 2. EFPNet with GT focus, meaning that we directly replace the output of FVP with GT error mask. 3. EFPNet with solid focus, we set focus scope to 0.3 and 0.7 respectively. As shown in below, there's still potential for NoC when GT is utilized, meaning that FIoU is not just a decorative metric but a crucial measurement for global effect rather than model capacity. In our final revision, We'll also provide a scatter-plot figure (FIoU-$\delta$IoU) for different focus mechanism, while first/subsequent click will be separately discussed.
> | Focus Mechanism | Berkeley NoC@90 | Berkeley FIoU | Berkeley 1-IoU | DAVIS NoC@90 | DAVIS FIoU | DAVIS 1-IoU | SBD NoC@90 | SBD FIoU | SBD 1-IoU |
> |--------------:|----------------:|--------------:|---------------:|-------------:|------------:|-------------:|-----------:|----------:|-----------:|
> | Predicted | 1.87 | 0.673 | 0.858 | 4.83 | 0.635 | 0.776 | 5.27 | 0.539 | 0.764|
> | GT | 1.72| 1.000  | 0.874 | 4.57| 1.000 | 0.799 | 4.96| 1.000  | 0.803 |
> | Solid (0.3)| 2.01| 0.146 | 0.833 | 5.15 | 0.109 | 0.761 | 5.43 | 0.221 | 0.739 |
> | Solid (0.7)| 1.95 | 0.335 | 0.834 | 5.21 | 0.298 | 0.754 | 5.49  | 0.204 | 0.732|
>
> Q4: Deep insight of FVP.
>
> A4: The key distinction is that FocalClick's difference map is a fixed heuristic that cannot adapt to different error patterns, while our FVP learns to predict errors from training data. This learned approach can capture complex error patterns that simple difference maps miss. Conceptually, we move the notion of “focus” from a fixed or post-processed object to a **learnable, transferable, and interaction-aware** latent variable that can be estimated from the current interaction context and reused throughout the refinement pipeline.

---

> > ### Author Rebuttal · Reviewer_msMH · 2026-04-01
> >
> > Thank you for your response. Your response resolved all my questions. I am willing to increase my score.

---

> > > ### Author Response · Authors · 2026-04-07
> > >
> > > Thank you for your approval to this paper！！

---

### Official Review · Reviewer_Gwct · 2026-03-11

**Soundness:** 3
**Presentation:** 3
**Significance:** 3
**Originality:** 4
**Overall Recommendation:** 5
**Confidence:** 4

**Summary:**

This paper improves over FCFI for click-based interactive segmentation. The method predicts an error mask over previous feedback, takes its max-connected component around the new click as the focus region, and refines feedback inside that region using image, feature and mask affinity. Corrected feedback is integrated via contrastive learning across focus regions and attention scoped to each focus. Evaluation on standarad benchmarks reports gains in NoC and related metrics.

**Compliance With Llm Reviewing Policy:**

Affirmed.

**Final Justification:**

The authors have adequately addressed my concerns, so I will maintain my score.

**Key Questions For Authors:**

Please refer to the weaknesses.

**Limitations:**

No. The paper does not contain a limitations section.

**Strengths And Weaknesses:**

Strengths

1. The paper is well written and technically sound. Although the architecture is complex, the overall flow is followable. Tables and figures are informative.

2. The work carries out very fine-grained optimization of interactive segmentation models and provides multi-faceted, multi-level improvements to the focus view design, offering some new insights.

3. Extensive experiments demonstrate that the proposed method achieves reasonable performance.

Weaknesses

1. Table 3 shows increased FLOPs compared to traditional schemes, and the computational efficiency is lower than schemes that pre-extract image features (e.g., SAM and InterFormer). This is constrained by the relatively complex pipeline of this work. More fundamental optimization to simplify the pipeline would be needed. The current approach is closer to complex engineering refinement.

2. GrabCut and Berkeley are relatively simple and outdated datasets. Although they are legacy evaluation standards, they should not serve as the main experimental data at this stage and could be moved to supplementary. High-resolution, high-precision segmentation data in the appendix (e.g., HQSeg44K) could be used as the main experiments instead.

3. Equations in the paper lack punctuation. Most equations should be followed by commas or periods where appropriate.

---

> ### Author Rebuttal · Authors · 2026-03-30
>
> Thanks for your approval to this paper!
>
> **Computational efficiency (R4-C1)**: We acknowledge the increased FLOPs shown in Table 3. The trade-off is one-stage simplicity vs two-stage computational cost. EFPNet avoids the two-stage overhead of methods like FocalClick, which requires 2× forward passes per interaction (global network + local refinement network). While our FLOPs are higher than FCFI's fixed-scope baseline, they remain lower than two-stage refinement methods.
>
> **Benchmark Modernity (R4-C2)**: we agree that GrabCut and Berkeley should not carry excessive weight as the primary evidence in 2026. They were retained because they remain common legacy benchmarks in interactive segmentation and allow direct comparison to a long line of prior work. In the revision, we will de-emphasize GrabCut and Berkeley in the main narrative and move more attention to the higher-resolution and higher-precision results included in the appendix.
>
> **Formula punctutation (R4-C3)**: This is a presentation issue on our side. We will carefully revise the manuscript so that displayed equations are followed by commas or periods where appropriate

---

> > ### Author Rebuttal · Reviewer_Gwct · 2026-04-03
> >
> > Thank you for your response.

---

> > > ### Author Response · Authors · 2026-04-07
> > >
> > > Thank you for your greatest approval to this paper！！

---

### Official Review · Reviewer_Uxsy · 2026-03-15

**Soundness:** 3
**Presentation:** 3
**Significance:** 2
**Originality:** 3
**Overall Recommendation:** 5
**Confidence:** 4

**Summary:**

The paper "Interactive Segmentation with Elaborate Focus Prior" introduces EFPNet (Elaborate Focus Prior Network), a one-stage framework designed to improve regional refinement in interactive segmentation. Interactive segmentation allows users to provide prompts, such as clicks, to guide a model in accurately localizing and segmenting specific objects.
Effective interactive segmentation requires high-quality "focus views", i.e., local windows around user clicks where pixels are revisited and optimized. Previous models like FCFI used a fixed focus scope, while others used slow, two-stage pipelines. These approaches often failed to capture the complex correlation between click positions and actual object geometry, leading to "detail leakage" or redundant computation.

The proposed architecture EFPNet is an end-to-end framework that dynamically adapts the focus view to the specific error region identified by a user's click. It consists of three primary modules:

- Focus View Predictor (FVP): Instead of a preset window, FVP predicts an "erroneous mask" highlighting incorrectly classified pixels from previous feedback. It then deduces a dynamic focus view based on the largest connected error region around the new click. In a single-click scenario, FVP acts as a global instance segmentation task.

- Triple Affinity Correction (TAC): This module refines previous feedback by considering three types of affinity: pixel affinity (low-level visual details like color), feature affinity (mid-level backbone features), and mask affinity (high-level semantic uncertainty).

- Clicked-with-Focus Integration (CFI): To ensure stable mask output and better use of focus priors, CFI utilizes Inter-Focus Contrastive Learning (FCL) for temporal consistency across historical clicks and Intra-Focus Attention (FA) to limit the network's perception to relevant local areas.

**Compliance With Llm Reviewing Policy:**

Affirmed.

**Final Justification:**

The rebuttal adressed my main concerns so I updated my recommendation.

**Key Questions For Authors:**

1. The Focus View Predictor (FVP) relies on a binary threshold $\tau$ to determine the extent of the focus region from the predicted erroneous mask. How sensitive is the final segmentation performance (NoC) to the choice of this threshold across different datasets? Specifically, does a threshold tuned for natural images (like DAVIS) generalize to datasets with significantly different object scales or textures?

2. The FVP deduces the focus region based on the max-connected component of the predicted error around a click. In scenarios with heavy occlusions or complex geometries (e.g., a bicycle partially hidden by a fence), the error region might be physically fragmented. How does EFPNet handle cases where the "max-connected component" only covers a small fraction of the actual required refinement area?

3. In the TAC module, you combine pixel, feature, and mask affinities. Could you provide a more granular ablation study (or quantitative data) showing the relative importance of $W_P$, $W_F$, and $W_M$ separately? Specifically, how much does the low-level pixel affinity ($W_P$) contribute to recovering thin structures versus the high-level semantic features ($W_F$)?

4. In the Clicked-with-Focus Integration (CFI) module, you use inter-focus contrastive learning to ensure stability. Does the model exhibit any "oscillation" or "forgetting" in very long interaction sequences (e.g., 20+ clicks)? Does the intra-focus attention ever lead to the model ignoring valid global context as more clicks are added?

**Limitations:**

While the authors briefly mention some failure cases in the qualitative analysis (e.g., regarding the precision of the Focus View Predictor), the paper lacks a dedicated, comprehensive section discussing limitations. Here are several limitations of the proposed method:

- Although the proposed method outperforms the existing SOTA in simple datasets, its effectivness declines in more complex datasets.

- The authors should discuss how the model behaves when the "max-connected component" logic of the FVP encounters physical occlusions. If an object is split into two visible parts by an overlaying obstacle, the current focus mechanism might struggle to refine both simultaneously.

- The reliance on specific thresholds ($\tau$) and loss weights ($\lambda$) should be addressed. A limitation would be the potential lack of robustness if these parameters need to be re-tuned for significantly different domains (e.g., satellite imagery vs. the natural images used in the benchmarks).

**Strengths And Weaknesses:**

**1. Soundness**

*Strengths:*

- The authors provide a clear mathematical formulation for their core modules, including the Focus View Predictor (FVP), Triple-Affinity Correction (TAC), and Clicked-with-Focus Integration (CFI).

- The paper evaluates the model on four standard benchmarks: GrabCut, Berkeley, DAVIS, and SBD. It uses established metrics such as Number of Clicks (NoC), Number of Failure (NoF), and Seconds Per Click (SPC).

- The results are compared against a wide range of state-of-the-art (SOTA) methods, including two-stage models like FocalClick and one-stage models like FCFI. EFPNet consistently shows performance gains (e.g., reaching 90% IoU on DAVIS with fewer clicks than previous methods).

- The authors also propose a new metric, i.e., FIoU, to  evaluate the performance and robustness of a model's Focus View Predictor (FVP).

*Weaknesses:*

- The FVP relies on a binary threshold ($\tau$) to control the focus view's range. While it's noted as a control mechanism, the sensitivity of the final segmentation to this specific parameter across different object types is not fully detailed.

- While the FVP is supervised by focal loss, the integration of multiple losses (segmentation loss, affinity loss, and contrastive loss) can be difficult to balance. More detail on how these losses were weighted during training would strengthen the soundness.

**2. Presentation**

*Strengths:*

- Figure 1 and Figure 2 effectively contrast EFPNet with the prior FCFI baseline and clearly illustrate the end-to-end framework.

- The paper clearly identifies four specific drawbacks of previous one-stage methods (static focus, no first-click refinement, incomplete affinity, and unstable integration) and addresses each with a dedicated module.

*Weaknesses:*

- Some sections, particularly the Intra-Focus Attention and the $log(\cdot)$ enforcement for attention values, are mathematically dense and could benefit from a more intuitive high-level explanation to improve readability for a broader audience.

- While generally clear, the use of multiple acronyms (FVP, TAC, CFI, FCL, FA) requires the reader to frequently cross-reference the text to follow the logic.

- In Table 1 and Table 2, there are some results that also need to be indicated in bold. For instance, in Table 1, MFP and RefCut also achieved 1.34 in GrabCut  NoC\%85, but they were indicatd in light colors.
Similarly, in Table 2, SegNext also achieved the best results, i.e., 0.967, along with the proposed method but not indicated in bold.

**3. Significance**

*Strengths:*

- The paper addresses a critical bottleneck in interactive segmentation: the high computational cost of two-stage refinement. By providing a SOTA one-stage framework, it makes high-quality interactive segmentation more feasible for real-time applications.

- The FVP's ability to handle the "first-click" as a full-object segmentation task  is a significant practical improvement over methods that require a secondary interaction round before local refinement begins.

*Weaknesses:*

- As shown in the experiment table, for simpler datasets like GrabCut, EFPNet marginally outperforms the existing methods, however, in more complex datasets like DAVIS, its success falls behind existing SOTA methods.

**4. Originality**

*Strengths:*

- Replacing the "fixed scope" focus used in FCFI with a learnable, dynamic Focus View Predictor based on an erroneous mask is a novel and intuitive approach to local refinement.

- Integrating pixel, feature, and mask-level affinities (TAC) to correct historical feedback is a notable contribution that acknowledges the limitations of using deep features alone.

*Weaknesses:*

- The authors explicitly state they "inherit the core idea of FCFI". While the technical improvements are meaningful and well-motivated, the work primarily builds upon an existing architectural framework rather than introducing a fundamentally new paradigm for interactive segmentation.

---

> ### Author Rebuttal · Authors · 2026-03-30
>
> Q1: The binary threshold $\tau$ for FVP.
>
> A1: This is a crucial concern since $\tau$ determines the elasticity of focus and is manually controllable. To achieve a fair comparison, **we use a shared threshold $\tau=0.5$ in both training and inference** for all experiment records in the main paper, without any model or datasets-specific tuning. Meanwhile, we're also interested in searching for the best $\tau$ for different datasets, where we perform a $\tau$-sweep evaluation ranging from 0.3~0.7 using four datasets. Our baseline model is trained with ViT-B backbone on COCO+LVIS. As is shown in the table below, all metrics--whatever NoC/1-IoU/FIoU, present a *U-style* or *Invert U-style* distribution around a respective $\tau$ values for each dataset, which **equals or slightly vary from 0.5 due to different object statistics**. For Berkeley/DAVIS filled with salient and mostly single-count object, their NoC metrics remain constant where $\tau \in \[0.45, 0.55\]$; while for SBD/HQSeg44K filled with multi-count texture-detailed objects, their optimal $\tau$ becomes 0.45 and 0.4. We argue that the **lack of focus details ($\tau$ is large)** is much more fatal than **content redundancy ($\tau$ is small)**, especially when it's difficult to precisely handle the erroneous region (See FIoU metric) for complex samples. You can also refer to the **abrupt decline** of 1-IoU metric when $\tau$ is lifted in zone $\[0.5, 0.7\]$, compared to **flat change** in $\[0.3, 0.5\]$. Meanwhile, we must be cautious not to set an extremely lower $\tau$ since the redundancy will overbalance which causes unstable output.
>
> | τ    | Berkeley NoC@90 | Berkeley FIoU | Berkeley 1-IoU | DAVIS NoC@90 | DAVIS FIoU | DAVIS 1-IoU | SBD NoC@90 | SBD FIoU | SBD 1-IoU | HQSeg44K NoC@90 | HQSeg44K FIoU | HQSeg44K 1-IoU |
> |------|------------------|---------------|----------------|--------------|------------|-------------|------------|----------|-----------|------------------|---------------|----------------|
> | 0.30  | 1.90  | 0.652 | 0.854 | 4.86 | 0.607 | 0.773 | 5.29 | 0.524 | 0.761 | 4.54 | 0.635 | 0.816 |
> | 0.40  | 1.88 | 0.664 | 0.854 | 4.85 | 0.622 | 0.775 | 5.27 | 0.531 | 0.763 | **4.51** | 0.638 | **0.823** |
> | 0.45 | **1.87** | 0.671 | 0.856 | **4.83** | 0.630 |**0.776** | **5.26** | 0.536 | **0.766** | 4.52 | 0.640 | **0.823** |
> | 0.50  | **1.87** | **0.673** | **0.858** | **4.83** | **0.635** | **0.776** | 5.27 | **0.539** | 0.764 | 4.52 | **0.641** | 0.822 |
> | 0.55 | **1.87** | **0.673** | 0.856 | 4.85 | 0.634 | 0.775 | 5.27 | 0.538 | 0.764 | 4.53 | 0.639 | 0.822 |
> | 0.60  | 1.89 | 0.669 | 0.852 | 4.86 | 0.629 | 0.773 | 5.29 | 0.533 | 0.762 | 4.55 | 0.636 | 0.819 |
> | 0.70  | 1.89 | 0.661 | 0.848 | 4.89 | 0.618 | 0.769 | 5.33 | 0.526 | 0.754 | 4.61 | 0.631 | 0.813 |
>
> Q2: Scenarios with complex apperance.
>
> A2: In fact, we concern about the design of **whole error prediction** instead of click-aware max-connected region, where multiple erroreous parts are simultaneously detected. However, we cannot guarantee the prediction stability since it has been converted to an **unbounded task** with no direct user guidance in isolated area, which causes ambiguity. Therefore, we observe that it's a robust choice to refine error regions *click-by-click* and *part-by-part*, to gradually form the result with **enhanced structural integrity**--When objects are partially split, users can click each part separately in subsequent interactions, while FVP will handle individually (please refer to the 3rd row of Figure 5 in main paper). It's also noteworthy that there're numerous samples with occlusion and complex geometries in **HQSeg44K** dataset, where we also achieve a competitive result (See Table 7) with the help of fragmented refinement scheme.
>
> Q3: About Clicked-with-focus Integration (CFI).
>
> A3: Firstly, the mask token in intra-focus attention (FA) is meant to adaptively fuse every click focus features and is not likely to ignore global context. Secondly, when viewing from another perspective, our inter-focus contrastive learning (FCL) resembles the idea of *hard negative mining*--By controlling threshold $\tau_p$ and $\tau_n$, FCL pays more attention when the feature of newly-added click acts abnormal with each historical click, thus constructing a discriminative feature trajectory w.r.t click property. This process further eliminates the oscillation phenomenon during long-sequence interation.
>
> Q5: About Triple-Affinity-Correction (TAC) module.
>
> A5: We have listed the impact for each weighting term in Table 5, where we found that none of these terms are absolutely prominent versus another two. The refinement of thin structure should also take account of many factors e.g textures, category and lighting conditions. When the object is salient in human-vision, pixel-level affinity is enough; otherwise we must rely on high-level semantics.
>
> Q6: Loss weights.
>
> A6: This is generic, where mask loss dominates and affinity/contrastive losses as auxiliary.

---

> > ### Author Rebuttal · Reviewer_Uxsy · 2026-04-03
> >
> > Thank you for your detailed response.

---

> > > ### Author Response · Authors · 2026-04-07
> > >
> > > Thank you for your positive opinion to this paper！！

---

### Decision · Program_Chairs · 2026-04-30

**Decision:**

Accept (regular)

**Comment:**

The paper received positive recommendations, with scores of 5, 5, 4, and 4. Reviewers found the paper to be well written and technically robust, supported by extensive experimental validations. The authors' rebuttal effectively addressed the reviewers' concerns regarding the binary threshold for FVP, the pseudo-view strategy, and the correlation between FIoU and NoC. All reviewers acknowledged the rebuttal as "fully resolved."
The area chair agrees with the reviewers' evaluations and recommends accepting this paper.